# Tixagevimab/Cilgavimab for COVID-19 Pre-Exposure Prophylaxis in Hematologic Patients—A Tailored Approach Based on SARS-CoV-2 Vaccine Response

**DOI:** 10.3390/vaccines12080871

**Published:** 2024-08-01

**Authors:** Krischan Braitsch, Samuel D. Jeske, Jacob Stroh, Maike Hefter, Louise Platen, Quirin Bachmann, Lutz Renders, Ulrike Protzer, Katharina S. Götze, Peter Herhaus, Mareike Verbeek, Christoph D. Spinner, Florian Bassermann, Marion Högner, Bernhard Haller, Jochen Schneider, Michael Heider

**Affiliations:** 1TUM School of Medicine and Health, Department of Internal Medicine III, University Medical Center, Technical University of Munich, 81675 Munich, Germanym.heider@tum.de (M.H.); 2TUM School of Medicine and Health, Institute of Virology, University Medical Center, Technical University of Munich, 81675 Munich, Germany; 3TUM School of Medicine and Health, Department of Nephrology, University Medical Center, Technical University of Munich, 81675 Munich, Germany; 4TUM School of Medicine and Health, Department of Internal Medicine II, University Medical Center, Technical University of Munich, 81675 Munich, Germany; 5Center for Translational Cancer Research (TranslaTUM), Technical University of Munich, 81675 Munich, Germany; 6German Consortium for Translational Cancer Research (DKTK) Partner Site TUM, German Cancer Research Center Heidelberg (DKFZ), 69120 Heidelberg, Germany; 7Bavarian Cancer Research Center (BZKF), 81675 Munich, Germany; 8TUM School of Medicine and Health, Institute of AI and Informatics in Medicine, University Medical Center, Technical University of Munich, 81675 Munich, Germany

**Keywords:** COVID-19, hematologic malignancy, tixagevimab/cilgavimab, neutralizing antibodies, PrEP

## Abstract

Patients with hematologic malignancies still face a significant risk of severe coronavirus disease 2019 (COVID-19). The severe acute respiratory syndrome coronavirus type 2 (SARS-CoV-2)-neutralizing monoclonal antibody combination tixagevimab/cilgavimab (TIX/CGB) could be administered to immunocompromised patients for pre-exposure prophylaxis (PrEP) before the emergence of TIX/CGB-resistant COVID-19 Omicron variants. TIX/CGB application could be carried out regardless of the host’s immune response to previous active SARS-CoV-2 vaccinations or infections. Because the efficacy of COVID-19 PrEP remains unclear, especially in SARS-CoV-2-seropositive patients, German national guidelines recommended TIX/CGB PrEP only for SARS-CoV-2-seronegative patients in addition to an intensified active vaccination schedule. Having followed these guidelines, we now report the characteristics and outcomes of 54 recipients of TIX/CGB PrEP in SARS-CoV-2-seronegative patients with hematological disease from a German tertiary medical center and compare them to 125 seropositive patients who did not receive any PrEP. While the number of patients with B-cell lymphomas was significantly higher in the seronegative cohort (33 (61%) vs. 18 (14%) cases, *p* < 0.01), patients with myeloid diseases were significantly more frequent in the seropositive cohort (51 (41%) vs. 5 (9%) cases, *p* < 0.01). Strikingly, patients who had undergone allogeneic hematopoietic stem cell transplantation were significantly more likely (forty-nine (39%) vs. six (11%) cases, *p* < 0.01) to be SARS-CoV-2 seropositive. We observed that prophylactic application of TIX/CGB PrEP to a highly vulnerable group of SARS-CoV-2-seronegative patients resulted in a similar number of COVID-19 breakthrough infections compared to the untreated seropositive control group (16 (32%) vs. 39 (36%), *p* = 0.62) and comparable COVID-19-related outcomes like hospitalization and oxygen requirement throughout an extended follow-up period of 12 months. In conclusion, our results support the tailored approach of administering TIX/CGB PrEP only to SARS-CoV-2-seronegative patients during the COVID-19 pandemic and might provide a rationale for similar strategies during future outbreaks/diseases, especially in times of initial limited availability and/or financial constraints.

## 1. Introduction

Patients with hematological malignancies who develop coronavirus disease (COVID-19) following infection with severe acute respiratory syndrome coronavirus 2 (SARS-CoV-2) have been shown to be at high risk for progression toward severe disease, hospitalization and necessity of intensive care unit (ICU) admission [1]. This has been attributed to a compromised humoral and cellular immune response, both due to the underlying hematologic malignancy itself as well as due to the administered antineoplastic therapy, including B-cell-targeting antibodies [2,3]. While the rapid availability of several messenger ribonucleic acid (mRNA)-based vaccines in early 2021 has been integral in preventing severe disease and reducing hospitalization and death in the majority of the population, the effectiveness of these vaccines in patients with hematological malignancies is limited due to disease and treatment-associated immune impairment [4]. Even though the emergence of newer SARS-CoV-2 variants has resulted in milder disease, lower hospitalization and death, studies in hematologic patients have demonstrated a continuous considerable morbidity and mortality associated with COVID-19 in this vulnerable population [5].

Tixagevimab/cilgavimab (TIX/CGB) is a long-lasting neutralizing monoclonal antibody combination of two Fc-modified human monoclonal antibodies that bind to two unique sites of the spike glycoprotein of SARS-CoV2, whose sequence was obtained from SARS-CoV-2 convalescent patients [6]. The intramuscular combination TIX/CGB was approved by the U.S. Food and Drug Administration (FDA) and European Medicines Agency (EMA) in December 2021 and March 2022 as a primary pre-exposure prophylaxis (PrEP) in immunocompromised patients. The approval was based on the PROVENT phase 3 trial, which showed a >80% relative risk reduction for symptomatic COVID-19 infection in patients receiving TIX/CGB; however, it only included a very limited number of patients with hematological malignancies and was conducted before the omicron variants became predominant, and, as a major limitation, it did not include patients who had been vaccinated or recovered from SARS-CoV-2 infection [7]. Despite limited data supporting the efficacy of TIX/CGB PrEP in seropositive patients, both FDA and EMA labels permitted broad administration of TIX/CGB PrEP to immunocompromised patients, regardless of their immune response to SARS-CoV-2 vaccines or previous COVID-19. In contrast, German national guidelines recommended TIX/CGB PrEP should only be given to patients who remained seronegative after full vaccination and/or infection. Seropositive patients were advised to follow intensified active vaccination schedules [8]. Here, we present real-life experience from a single-center German tertiary university hospital, where only seronegative SARS-CoV-2 vaccine non-responders received TIX/CGB and compare characteristics and outcomes to a SARS-CoV-2-seropositive cohort of hematological patients who did not receive TIX/CGB and provide data from an extended follow-up period of 6 and 12 months.

## 2. Materials and Methods

### 2.1. Study Design

In this retrospective single-center cohort study, we evaluated characteristics and COVID-19-related outcomes (breakthrough infections, necessity of hospitalization and/or oxygen support) of patients with hematological malignancies who were undergoing/had undergone treatment or were considered immunocompromised based on their hematologic disease. All patients were receiving medical care at the Department of Hematology/Oncology at the University Hospital rechts der Isar, Technical University of Munich. Patients with hematologic disease who were receiving immunocompromising therapy, were undergoing follow-up, or were considered immunocompromised due to their underlying disease were screened for eligibility to receive TIX/CGB PrEP based on their individual response to the recommended SARS-CoV-2 vaccination regimes, as indicated per national recommendation [9,10]. Response to vaccination was assessed by determining anti-S neutralizing antibodies (nAB). Patients with a test-related nAB titer cut off <50 AU/mL were considered to be seronegative vaccine non-responders or poor responders. Per German national guidelines, these patients received a recommendation to receive TIX/CGB PrEP [8]. Patients were screened for nAB titers from December 2021 to December 2022, and eligible patients received TIX/CGB PrEP from March 2022 to December 2022, when both the EMA and FDA altered their approval of TIX/CGB based on studies that suggested that the emerging omicron variants BQ1.1 and XBB were no longer sensitive to various monoclonal antibody combinations including TIX/CGB [11,12].

Patients with nAB titers > 50 AU/mL were considered seropositive, did not receive TIX/CGB following national guidelines, were encouraged to follow an intensified active vaccination schedule and served as a control cohort. All patients were followed up for 12 months. Baseline characteristics, co-morbidities and outcome-relevant parameters were assessed during routine in-/outpatient visits and medical chart reviews. TIX/CGB PrEP-associated tolerability issues were also evaluated. Patients who had received therapeutic neutralizing antibodies such as sotrovimab for a SARS-CoV-2 infection were excluded from the analysis. Immunologic events were defined as number of vaccinations and documented previous infections.

### 2.2. Neutralizing Antibody Measurements

SARS-CoV-2 nAB in patients’ sera was measured using an iFlash 1800 platform (YHLO Biotechnology, Shenzhen, China). Specifically, we used the iFlash 2019-nCoV nAB CLIA based on the competition between serum antibodies and recombinant angiotensin-converting enzyme 2 to bind to the SARS-CoV-2 Wuhan strain receptor-binding domain [13]. According to the manufacturer’s instructions, this surrogate neutralization assay was adapted for quantification, and the upper limit of nAB value was 800 AU/mL. Results < 10 AU/mL were not further quantified but were reported as negative. Patients with negative results were assigned a value of 0 AU/mL for calculation of median nABs in the PrEP cohort.

### 2.3. Ethical Approval

This study was conducted according to the principles of the Declaration of Helsinki. It was approved by the Medical Ethics Committee of the Technical University of Munich (approval number 163/21 S-SR 423/17 S). All patients receiving/having received immunocompromising therapy at our center were assessed. All participants enrolled in this retrospective follow-up study provided informed consent.

### 2.4. Statistical Analysis

Statistical tests were performed using GraphPad Prism (Version 10.2.3). *p*-values < 0.05 were considered statistically significant. Distributions of infections over time were estimated using the Kaplan–Meier method and compared using the log-rank (Mantel–Cox) test. Cox proportional hazard regression was used to assess the impact of various covariates on breakthrough infections, and hazard ratio (HR) and its 95% confidence interval (CI) were reported to quantify the effect size of each covariate. Chi-squared test was used to compare categorical data between two samples. Continuous variables were compared using the *t*-test. Matched pair analysis considering patients with the same values for anti-CD20 treatment, anti-CD38 treatment, autoSCT, alloSCT and age ≥ 65 was conducted using R (Version 4.4.1). Due to the retrospective analysis, sample size was chosen based on the number of available patients in each cohort during the observed time frame.

## 3. Results

Following German national guidelines, all patients were tested for SARS-CoV-2 nAB before receiving TIX/CGB PrEP [8]. Patients with nAB titers < 50 AU/mL who failed to respond to previous vaccinations or build up humoral immunity following COVID-19 were considered seronegative and were advised to receive TIX/CGB PrEP.

We identified 61 patients with hematologic disease who received TIX/CGB PrEP, of which 54 patients met the inclusion criteria. The seven excluded patients either received TIX/CGB as treatment during an active SARS-CoV-2 infection or had a titer of >50 AU/mL. The 54 included patients received TIX/CGB 150 mg/150 mg once according to EMAs March 2022 label.

A total of 178 patients with hematologic disease who underwent a nAB screening but did not receive TIX/CGB PrEP were screened. Fifty-three of these patients who either had received therapeutic mAB treatment with sotrovimab due to a SARS-CoV-2 infection or had a nAB titer of <50 AU/mL and did not receive TIX/CGB PrEP due to patients or physicians’ choice were excluded from this analysis. The remaining 125 patients with hematologic disease were considered seropositive and therefore included in the control cohort (Figure 1).

Notably, most patients who failed to respond to vaccination had lymphoid malignancy (*n* = 48, 89%), predominantly B-cell lymphoma (*n* = 33, 61%) or plasma cell dyscrasia (*n* = 13, 24%). Patients with plasma cell dyscrasia were more common in the control group, but this difference was not statistically significant (45 (36%) vs. 13 (24%), *p =* 0.12). However, there were significantly more B-cell lymphoma patients in the TIX/CGB PrEP cohort (33 (61%) vs. 18 (14%), *p* < 0.01) and significantly more patients with myeloid diseases such as acute myeloid leukemia (AML), myelodysplastic syndrome (MDS) or myeloproliferative neoplasms (MPN) in the seropositive control cohort (51 (41%) vs. 5 (9%), *p* < 0.01) (Table 1). The median age at the time of nAB screening was significantly lower in the control cohort with 59 (range 23–94) years compared to 70 (26–88) years in the group that received TIX/CGB PrEP (*p* < 0.01) (Table 1).

In both cohorts, most patients were undergoing or had previously received chemotherapy or other targeted therapies. In line with the difference in underlying diseases, the rate of anti-CD20 therapy was significantly higher in the TIX/CGB PrEP group (32 (59%) vs. 22 (18%), *p* < 0.01), while there was no difference for anti-CD38 treatment (8 (15%) vs. 17 (14%), *p =* 0.83). The number of patients who had received allogeneic SCT was significantly higher in the control cohort (49 (39%) vs. 6 (11%), *p* < 0.01), while the rates of autologous SCT showed no relevant difference (19 (35%) vs. 41 (33%), *p =* 0.76) (Table 1).

Median nAB titers in the TIX/CGB PrEP cohort were 0 AU/mL (0–40), and patients had 3 (2–5) immunological events before nAB measurement. Only five patients (9%) reported or had a documented previous SARS-CoV-2 infection (data available for 53 patients), and patients had received an average of 3.45 (0–5) vaccinations before nAB measurement. In the seropositive control group, median nAB titers were 722 AU/mL (range 52–800), with 800 AU/mL being the reported maximum of the assay. Forty-two patients had reported a previous COVID-19 infection, which was significantly more than in the TIX/CGB PrEP group (*p* < 0.01), while the number of previous vaccinations was significantly lower compared to the PrEP cohort, with an average of 2.96 (0–5) (*p* < 0.01). While the number of previous infections and vaccinations differed between the two groups, the composite parameter immunologic events did not show a significant difference (3 (2–5) vs. 3 (0–6), *p =* 0.16).

For those patients with available vaccination dates (28/54 and 31/125), the time between the last vaccination and nAB measurement was not significantly different between the groups (108 (28–358) vs. 131 days (9–316)), *p =* 0.14) (Table 2).

In the median, the time between nAB assessment and TIX/CGB injection was 7 (0–143) days, and TIX/CGB was administered a median of 116 days (6–368; data available for 28/54 patients) after the last vaccination. TIX/CGB was well tolerated; one patient reported minor local bleeding and pain at the gluteal injection site, *n* = 52 had no injection-related side effects, and one patient was unavailable to provide tolerability data. Median platelet count was 162 G/L (13–336) on the day of injection, and coagulation parameters were available for 23 patients with a median INR of 1.0 (0.9–1.0).

To allow for better comparability between both groups, breakthrough infections were defined as SARS-CoV-2 infections that occurred after the date of initial nAB measurement, which was used to determine eligibility for TIX/CGB PrEP. After 3 and 6 months of follow-up, three (6%) and eight patients (17%) in the TIX/CGB PrEP and eleven (9%) and twenty-three (19%) patients in the control cohort had breakthrough infections (percentage calculated using Kaplan–Meier estimates). After an extended follow-up of one year, breakthrough infections were observed in 16 (32%) and 39 (36%) patients in the TIX/CGB PrEP and control cohorts, respectively. The rates of breakthrough infections are displayed in Figure 2; no significant differences were seen during the 12-month follow-up (*p* = 0.62) (Figure 2). In individuals who experienced breakthrough infections, the infection occurred a median of 140 days (31–324) after TIX/CGB. Among patients with a SARS-CoV-2 breakthrough infection, a total of six out of sixteen (37%) patients were hospitalized in the TIX/CGB PrEP group compared to nine out of thirty-seven (24%, data unavailable for two patients) in the control cohort (*p =* 0.19). Oxygen support was required in one (6%) and four (11%, data unavailable for five patients) patients in the TIX/CGB PrEP group and control group, respectively (*p =* 0.56).

Multivariate analysis conducted for all patients showed no significant impact of TIX/CGB PrEP (HR 0.92; 95% CI 0.45–1.80, *p =* 0.81) on the rate of SARS-CoV-2 breakthrough infections. A significantly higher risk for breakthrough infections was observed in patients treated with anti-CD38 mABs (HR 2.55; 95% CI 1.18–5.24, *p* = 0.01), while no significant association was seen for the parameters anti-CD20 mAB treatment, age ≥ 65 or auto/alloSCT (Figure 3). In addition, exact matching resulted in a total of *n* = 66 matched patients (*n* = 33 in each group), which showed no significant difference in breakthrough infections after 12 months (*p =* 0.78) (Table A1, Figure A1).

Mortality of any cause was similar in both groups; seven (13%) patients in the TIX/CGB PrEP cohort and eighteen (14%) patients in the control cohort died during the 12-month follow-up (*p* = 0.80). In the TIX/CGB PrEP cohort, two deaths were considered to be related to COVID-19; other causes of death were ARDS with fungal pneumonia (*n* = 2), cardiac (*n* = 1) and sepsis (*n* = 1), and one patient died due to cancer progression. In the seropositive control group, two deaths were considered to be related to COVID-19; other causes were pneumonia (*n* = 3), bleeding (*n* = 1), sepsis (*n* = 3) and GvHD (*n* = 2), and seven patients died from cancer progression.

## 4. Discussion

While the global disease burden of COVID-19 is decreasing due to vaccine- and SARS-CoV-2 exposure-induced immunity, immunocompromised patients still face a significant risk of severe disease outcomes owing to their diminished protective immune response [14]. Intensified vaccine schedules in such populations, as recommended by German national guidelines, for example, can prompt a robust humoral immune response in many of these patients [9,10]. However, for an extremely vulnerable group of seronegative patients, PrEP with mABs like TIX/CGB presents an attractive additional protective option.

Both FDA and EMA labels permitted broad administration of TIX/CGB to immunocompromised patients, irrespective of their response to vaccination or previous COVID-19, based on the PROVENT 3 study [7]. However, the effect of such a drug combination in patients with positive nAB status following the recommended vaccination schemes or prior SARS-CoV-2 exposure remains unclear, especially since vaccination or previous COVID-19 had been the main exclusion criteria in the PROVENT 3 trial. Furthermore, recent studies suggest an altered immune response to vaccines in patients who had previously received mABs [15,16]. While the exact degree of this interference remains to be determined, some regulatory agencies recommended deferring vaccination after mAB exposure for a certain period of time. In the context of TIX/CGB PrEP, this could mean both reduced efficacy and/or unnecessary delays in vaccine dispensing in seropositive patients. Lastly, limited financial resources and slow rollout of novel mABs due to manufacturing and distribution challenges during a crisis such as the COVID-19 pandemic might initially lead to insufficient availability for a large number of immunocompromised patients. For all the above reasons, a more tailored approach to administering these and future other mABs seems warranted.

The German national guidelines therefore recommended TIX/CGB PrEP only for patients who remained seronegative after full vaccination and/or infection [8]. Our real-world study in a hematologic patient population, which was treated following these guidelines, showed that administration of TIX/CGB PrEP to this highly vulnerable group of seronegative patients resulted in a similar rate of SARS-CoV-2 breakthrough infections and COVID-19 severity compared to a seropositive control group. This observation was conserved in an exact matching model and confirmed in a multivariate analysis, which showed no significant association between TIX/CGB PrEP administration and the rate of breakthrough infections. While both the EMA and FDA have since altered the approval status for the use of TIX/CGB PrEP based on a lack of efficacy against current variants [11,12], our data support such a tailored SARS-CoV-2 nAB status-based approach for TIX/CGB PrEP, a strategy that should be further validated in prospective trials during upcoming outbreaks/diseases.

Intriguingly, in this real-world observational study, the majority of patients with myeloid disease and patients who had undergone allogeneic HSCT showed humoral responses to SARS-CoV-2 vaccines. Patients after allogeneic HSCT, with or without GvHD, are commonly considered to be at very high risk for COVID-19 and other viral respiratory infections due to prolonged immunosuppression [17,18]. In fact, in our analyzed general hematologic patient population, only 30% of patients were considered seronegative and therefore received TIX/CGB PrEP. Previous work identified 35.2% and 37.02% of patients in a similar cohort to exhibit undetectable levels of anti-S and anti-RBD IgG antibodies, while another study on vaccine efficiency in patients after allogeneic SCT showed 57% of patients developed a humoral response to SARS-CoV-2 vaccines, which is consistent with our data [19,20].

Our results are also in line with recent data from Angotzi and colleagues, who reported similar outcomes from a smaller cohort of seronegative patients treated with TIX/CGB PrEP compared to a seropositive control cohort [21]. Furthermore, several other non-controlled studies with shorter follow-up periods have shown similar rates of SARS-CoV-2 infections. In a large Czech multicenter retrospective analysis, Demel and colleagues reported that 16% of hematologic TIX/CGB recipients had contracted a SARS-CoV-2 infection with a hospitalization rate of 25% after 6 months [22]. Another study with a median follow-up of 151 days observed a 9.3% incidence of SARS-CoV-2 infection in TIX/CGB PrEP-treated hematologic patients with only one patient (0.5%) requiring hospitalization, which might have been due to a lower incidence of SARS-CoV-2 infections in the general population during this time [23]. Initial pharmacokinetic studies of TIX/CGB suggest an effective concentration of up to 12 months and therefore warrant the longer observation period so far exclusively reported in our study [24].

The higher risk of SARS-CoV-2 infection in patients undergoing anti-CD38-based therapies observed in our multivariate analysis (HR 2.55; 95% CI. 1.18–5.24; *p =* 0.01) is in agreement with the observation that patients with plasma cell dyscrasia, especially those treated with anti-CD38 agents, are at high risk for breakthrough infections due to underlying immune deficiency from the disease itself and a suboptimal antibody response following vaccination [25,26]

Several studies that have investigated TIX/CGB as a treatment for symptomatic COVID-19 in various populations, including some studies in immunocompromised patients, have shown conflicting results with no or modest efficacy of this antibody combination in the therapeutic setting [27,28,29,30,31]. Another group has studied the effect of TIX/CGB as post-exposure prophylaxis in at-risk patients with high exposure but was unable to meet the primary efficacy endpoint of post-exposure prevention of symptomatic COVID-19 [32]. Some of these studies might have struggled due to the emergence of tixagevimab/cilgavimab-insensitive variants. However, it is well conceivable that the effectiveness of such treatment or post-exposure prophylaxis would be significantly higher in a selected population of seronegative immunocompromised patients and therefore warrants further investigation.

The safety data observed in our study, with only one patient (1.8%) in the TIX/CGB cohort experiencing local bleeding and pain after injection, is in line with the low rate of adverse events in the original phase III trial, where 2.4% experienced injection site reactions [7].

Our study has several limitations. First, it is a single-center retrospective observational study of two distinct non-randomized patient groups. However, withholding TIX/CGB PrEP from seronegative immunocompromised patients would have been unethical and was therefore not considered. Given the retrospective nature of the study, exact vaccination dates could not be determined for a significant proportion of the patients in both cohorts. Second, despite the FDA’s recommendation to apply higher doses of 300/300 mg, we followed European guidelines as well as the EMA approval and continued to administer a dose of 150/150 mg to all patients at our hospital. Lastly, despite the emergence of novel tixagevimab/cilgavimab-insensitive variants throughout our observational period, we were unable to confirm which variants caused breakthrough infections in our patient cohorts.

## 5. Conclusions

In summary, we demonstrate that prophylactic application of TIX/CGB PrEP in a highly vulnerable group of SARS-CoV-2-seronegative patients with hematologic malignancies resulted in similar COVID-19-related outcomes compared to an untreated seropositive control group. Our results support the SARS-CoV-2 nAB status-based tailored approach of administering TIX/CGB PrEP to seronegative patients. This strategy needs to be further evaluated in randomized trials and should be considered in the investigation of novel monoclonal antibodies such as AZD3152 (phase I/III trial SUPERNOVA, NCT05648110) or Pemivibart (VYD222, phase III trial CANOPY, NCT06039449), the latter of which has just recently been granted a broad emergency use authorization by the FDA [33].

## Figures and Tables

**Figure 1 vaccines-12-00871-f001:**
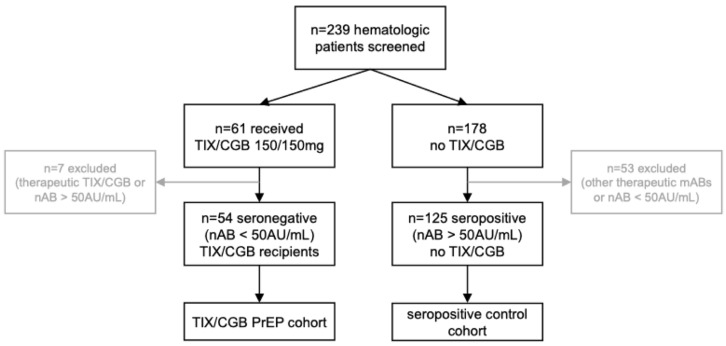
Flow diagram of study design and patient screening process.

**Figure 2 vaccines-12-00871-f002:**
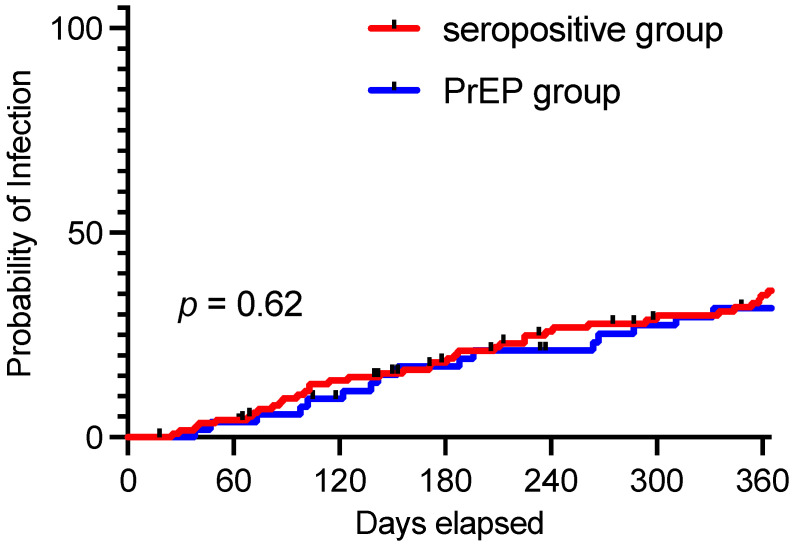
SARS-CoV-2 breakthrough infection rates 1 year after nAB measurement. Kaplan–Meier curve showing infections over time during the 1-year follow-up period for each group. PrEP = pre-exposure prophylaxis.

**Figure 3 vaccines-12-00871-f003:**
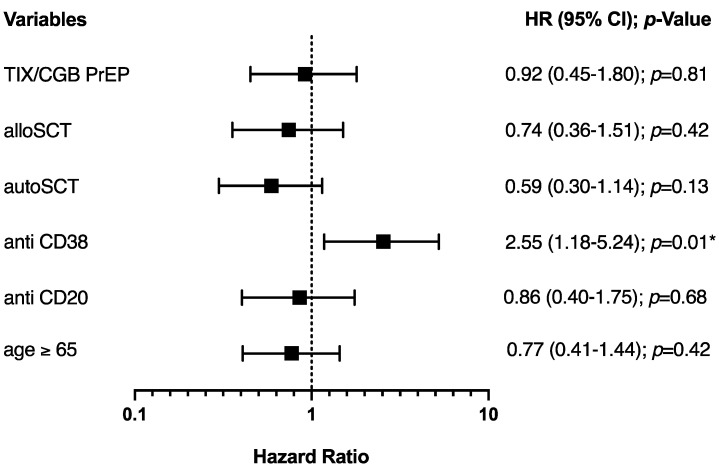
Multivariate analysis for 1-year SARS-CoV-2 breakthrough infections for all patients. Forest plot of multivariate analysis showing hazard ratios (HR) and 95% confidence intervals (CI) between breakthrough infections and indicated variables on the left. * *p* < 0.05.

**Table 1 vaccines-12-00871-t001:** Patient characteristics. Baseline characteristics are listed for both TIX/CGB PrEP and seropositive control groups. Relevant *p*-values are provided as indicated. Other disease entities were paroxysmal nocturnal hemoglobinuria, aplastic anemia and blastic plasmacytoid dendritic cell neoplasm. NHL = non-Hodgkin’s lymphoma; ALL = acute lymphocytic leukemia; AML = acute myeloid leukemia; MDS = myelodysplastic syndrome; MPN = myeloproliferative neoplasms; CAR-T = chimeric antigen receptor T cells. Antineoplastic therapy included but was not limited to the agents/modalities listed below.

	TIX/CGB PrEP	Seropositive	*p*-Value
**number of patients**	54	125	
**age, years**			
median (range)	70 (26–88)	59 (23–94)	<0.01
≥65 years	35 (65%)	39 (31%)	
**sex**			
male	35 (65%)	87 (70%)	0.53
female	19 (35%)	38 (30%)	
**disease**			
plasma cell dyscrasia	13 (24%)	45 (36%)	0.12
B-NHL	33 (61%)	18 (14%)	<0.01
T-NHL	2 (4%)	2 (2%)	
ALL	0 (0%)	3 (2%)	
AML, MDS	3 (6%)	43 (34%)	<0.01
MPN	2 (4%)	8 (6%)	
other	1 (2%)	6 (5%)	
**Previous therapy**			
cancer therapy	51 (94%)	117 (96%)	
active treatment during nAB measurement	33 (61%)	92 (74%)	0.10
anti-CD20	32 (59%)	22 (18%)	<0.01
anti-CD38	8 (15%)	17 (14%)	0.83
autoSCT	19 (35%)	41 (33%)	0.76
alloSCT	6 (11%)	49 (39%)	<0.01
CAR-T	3 (5%)	3 (2%)	
**COVID-19 history**			
Immunologic events; median (range)	3 (2-5)	3 (0–6)	0.16
Previous infection, *n*	5 (9%)	42 (34%)	<0.01
Previous vaccinations, mean (range)	3.45 (0–5)	2.96 (0–5)	<0.01
Serum nAB levels; median (range)	0 (0–40)	722 (52–800)	<0.01

**Table 2 vaccines-12-00871-t002:** Characteristics of breakthrough SARS-CoV-2 infections and additional risk factors. Parameters for breakthrough infections and additional risk factors are listed for both TIX/CGB PrEP and seropositive control groups. Relevant *p*-values are provided as indicated. COPD = chronic obstructive pulmonary disease. * from time of nAB measurement, % calculated from Kaplan–Meier estimates; ** based on patients with breakthrough infections.

	TIX/CGB PrEP	Seropositive	*p*-Value
**number of patients**	54	125	
**Timeline dates**			
Vaccination to nAB measurement, d (median, range)	108 (28–358)	131 (9–316)	0.14
nAB measurement to tixagevimab/cilgavimab, d (median, range)	7 (0-143)		
nAB measurement to infection, d (median, range)	148 (38–332)	171 (25–363)	0.94
**Breakthrough Infections**			
Month 3 *	3 (6%)	11 (9%)	0.62
Month 6 *	8 (17%)	23 (19%)	0.62
Month 12 *	16 (32%)	39 (36%)	0.62
Hospitalization **	6 (37%)	9 (25%)	0.19
Oxygen support **	1 (6%)	4 (11%)	0.56
**Risk factors**			
Hypertension	24 (44%)	34 (27%)	
Diabetes	9 (17%)	13 (10%)	
Congestive heart failure	4 (7%)	5 (4%)	
Arrhythmia	11 (20%)	11 (9%)	
Coronary artery disease	6 (11%)	6 (5%)	
Chronic kidney disease	4 (7%)	8 (6%)	
Asthma/COPD	0 (0%)	8 (6%)	
**Death during follow-up**	7 (13%)	18 (14%)	0.80
COVID-19 related	2 (29%)	2 (11%)	

## Data Availability

The original contributions presented in this study are included in the article. Further inquiries can be directed to the corresponding author/s.

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
