# Peer review of "Tixagevimab/Cilgavimab for COVID-19 Pre-Exposure Prophylaxis in Hematologic Patients—A Tailored Approach Based on SARS-CoV-2 Vaccine Response"

_vaccines, 2024, doi:10.3390/vaccines12080871_

Round 1
Reviewer 1 Report
Comments and Suggestions for Authors
As per the below file.

Reviewer 2 Report
Comments and Suggestions for Authors
In this paper, the authors have analysed the interest of a pre-exposure prophylaxis (PrEP) by the combination of tixagevimab/cilgavimab (TIX/CGB) in a series of 54 patients with hematologic malignancies who had no antibodies against SARS-CoV-2 despite several vaccinations and/or infection. They compare the occurence of a SARS-CoV-2 infection in this group of seronegative patients with PrEP to a group of 152 patients with hematological maligancies who have neutralizing antibodies against SARS-CoV-2; They observe a similar frequency of COVID-19 in both group of patients and conclude that the German strategy to use PrEP only in seronegative patients is efficient.
Major comments:;
- The authors should justify why they choose a one year follow-up. Because of the half-life of immunoglobulins, we could speculate that the protective effect of PrEP does not last more than 3 months. The auhors indicate p 7, line 223, that during the first 3 months 5 patients experienced COVID-19. They should indicate how many had COVID-19 in the seropositive group and provide statistical data.
- The test to detect neutralizing antibodies is not largely used and references concerning its characteristics should be indicated. How was the cut-off of 50 AU/mL determined to indicate that patients had no protective antibodies. Additionnaly, Furthermore, for very low values, the sensitivity of the tests is poor, and biology laboratories respond "less than ...."; under these conditions, how can the authors report a median value of 0 in the PrEP group?
- Lastly, some data are missing. for example, the authors indicate that they have available vaccination dates dor 24/58 and 31/125 patients. Missing data are a limitation to the resulst of the study, and this should be indicated.
Minor comments:
- page 3, line 102: units of antibodies are expressed as "AU/ML", line 109 or 124 "AU/ml". The international abbreviation is "mL".
- page 7, line 216: "percentage" does not require a capital character
Reviewer 3 Report
Comments and Suggestions for Authors
This is an interesting paper describing the German experience of tailoring prophylaxis against COVID9 infection in immunocompromised patients with haematological malignancies based on results of Vaccine response.
The study is well written but some improvements are suggested below:
1. Presentation of study design- a figure or flow diagram will make it easier to follow
2. Which vaccines are administered to patients? Do the type of vaccines affect the response?
3. Table 1 - further details will be helpful for example previous lines of therapy (this will help highlight if heavily treated patients are more likely to have inadequate response to vaccination, not just their type of haematological disease); also if they are on active treatment at the time of prophylaxis (?less likely to be efficacious); other factors such as the presence of hypogammaglobulinemia
For the same table, the description of previous therapy can also be expanded for improved clarifyt- cancer therapy is a very broad term- does this encompass CD20/CD38 abs as well as everything on the list or is this referring to other therapies ? How are the patients who have both antiCD38 and autoSCT, CAR-T included (are they included thrice which may influence the stats?)
It would certainly be interesting to know the timing of vaccinations prior to nAB measurement but we are aware that this is unfortunately unavailable for most of the patients for both groups as this would potentially have affected the response to vaccine especially if patients are on active therapy at the time of vaccinations.
It is intriguing to learn that patients who had undergone alloHSCT demonstrated humoral responses to COVID-19 vaccines- in this population, it would be interesting to know if the use of ongoing immunosuppression eg for GVHD affects the vaccine response and rate of infections, as this is certainly the group that causes the most anxiety.
Overall, I find this study to be well written but provision of further details if available can strengthen it and make it more informative.
